# Natalizumab Induces Changes of Cerebrospinal Fluid Measures in Multiple Sclerosis

**DOI:** 10.3390/diagnostics11122230

**Published:** 2021-11-29

**Authors:** Ranjani Ganapathy Subramanian, Dana Horakova, Manuela Vaneckova, Balazs Lorincz, Jan Krasensky, Eva Kubala Havrdova, Tomas Uher

**Affiliations:** 1Department of Neurology and Center of Clinical Neuroscience, First Faculty of Medicine and General University Hospital, Charles University in Prague, 128 21 Prague, Czech Republic; gs.ranjani9@gmail.com (R.G.S.); dana.horakova@vfn.cz (D.H.); lorinczbalazsneurologia@gmail.com (B.L.); eva.kubalahavrdova@vfn.cz (E.K.H.); 2Department of Radiology, First Faculty of Medicine, Charles University and General University Hospital in Prague, Katerinska 30, 128 08 Prague, Czech Republic; Manuela.Vaneckova@vfn.cz (M.V.); Jan.Krasensky@vfn.cz (J.K.)

**Keywords:** natalizumab, cerebrospinal fluid, brain atrophy, oligoclonal bands, MRI, multiple sclerosis

## Abstract

Background: There is a lack of knowledge about the evolution of cerebrospinal fluid (CSF) markers in multiple sclerosis (MS) patients undergoing natalizumab treatment. Aim: We aimed to evaluate the effect of natalizumab on basic inflammatory CSF and MRI measures. Methods: Together, 411 patients were screened for eligibility and 93 subjects with ≥2 CSF examinations ≤6 months before and ≥12 months after natalizumab initiation were recruited. The effect of natalizumab on CSF as well as clinical and paraclinical measures was analyzed using adjusted mixed models. Results: Natalizumab induced a decrease in CSF leukocytes (*p* < 1 × 10^−15^), CSF protein (*p* = 0.00007), the albumin quotient (*p* = 0.007), the IgG quotient (*p* = 6 × 10^−15^), the IgM quotient (*p* = 0.0002), the IgG index (*p* = 0.0004), the IgM index (*p* = 0.003) and the number of CSF-restricted oligoclonal bands (OCBs) (*p* = 0.0005). CSF-restricted OCBs positivity dropped from 94.6% to 86% but 26 patients (28%) had an increased number of OCBs at the follow-up. The baseline to follow-up EDSS and T2-LV were stable; a decrease in the relapse rate was consistent with a decrease in the CSF inflammatory markers and previous knowledge about the effectiveness of natalizumab. The average annualized brain volume loss during the follow-up was −0.50% (IQR = −0.96, −0.16) and was predicted by the baseline IgM index (B = −0.37; *p* = 0.003). Conclusions: Natalizumab is associated with a reduction of basic CSF inflammatory measures supporting its strong anti-inflammatory properties. The IgM index at the baseline predicted future brain volume loss during the course of natalizumab treatment.

## 1. Introduction

Multiple sclerosis (MS) is a chronic, demyelinating autoimmune disease. It is multifactorial in nature due to an interaction between environmental factors and an underlying genetic susceptibility. The most common clinical course of MS is relapsing-remitting MS (RRMS). There have been great advances in the treatment of RRMS after the discovery of monoclonal antibodies and other disease modifying therapies (DMT). Among them, natalizumab (NTZ) is a humanized recombinant antibody against α4-integrin and is highly effective in reducing disability progression, relapses and radiological disease activity in RRMS [1,2,3,4]. It acts primarily by inhibiting the interaction between very late activating antigen 4 (VLA-4) (expressed on all leukocytes except neutrophils) and vascular cell adhesion molecule 1 (VCAM-1) (expressed on endothelial cells) and thus inhibits leukocyte adhesion and migration across the blood-brain barrier (BBB) into the central nervous system (CNS) and to other organs, resulting in an increase in most immune cell subpopulations in the blood [5].

The binding of NTZ is more marked for B cells than for T cells, leading to a disproportionate increase in B cells in the blood following treatment [6]. Blood markers in response to NTZ treatment include an increase in the circulating hematopoietic stem and associated progenitor cells [7], a decrease of the CD4/CD8 ratio [8] and lymphocytosis [9,10]. Although there are consistent data documenting the efficacy of NTZ on immunological and biochemical changes in the blood, only a few studies have investigated NTZ-induced changes in the CSF where its anti-inflammatory effect can be evidenced by the decrease of CSF leukocyte counts, CD4+ and CD8+ T cells, CD19+ B cells and CD138+ plasma cells [11]. Additionally, NTZ normalizes the levels of B cells, monocytes, natural killer cells and dendritic cells in the CSF [5]. NTZ treatment is also associated with a reduction of CSF plasmablasts [12], especially in short-lived cells of the CNS compartment and, to a lesser extent, locally persisting long-lived plasma cells [13]. NTZ decreases the total CSF IgG [14,15,16] and IgM levels [15,16] as well as the IgG index [16] and intrathecally produced IgG [14]. Despite several studies that have reported the disappearance of CSF-restricted oligoclonal bands (OCBs) in a great proportion of patients [17], more recent research has found a persistence of OCBs in the vast majority of patients and a disappearance in only a few patients [13,14,15]. Interestingly, a direct comparison of OCB patterns before and after NTZ showed the persistence of individual bands in the CSF [13]. In addition, treatment with NTZ is associated with a decrease of CD5 cells and TNFRSF9 receptors as well as IL-12B, IL-18R1 and CXCL10 cytokines in the CSF [18]. All these results indicate the strong immunomodulatory effects of NTZ (1-new), which—besides modulating the T cell migration—dominantly affects the B cell compartment by inhibiting the anti-a4b1 B cell migration across the BBB (2-new) by participating in the activation of human memory B cells (3-new) and by supporting the long-lived plasma cells to remain in their survival niche (4-new). All of these aspects result in the prevention of migration of the immune cells into the CNS and neuroinflammation [19,20,21,22].

Considering that only a few small sample size studies have investigated the effects of NTZ on CSF measures, long-term longitudinal studies are needed to better understand the dynamics of the anti-inflammatory effect of NTZ in MS. Additionally, the association between NTZ-induced changes in the CSF and radiological disease activity on MRI has not been studied in detail. We hypothesized a strong effect of NTZ on the reduction of basic inflammatory CSF measures in RRMS patients. Conversely, the null hypothesis of the study was that there was no statistically significant relationship between NTZ treatment initiation and the reduction of inflammatory CSF measures.

## 2. Methods

### 2.1. Study Population

In this longitudinal retrospective study, we included relapsing-remitting MS patients treated with NTZ at the General University Hospital in Prague, Czech Republic. The inclusion criteria were as follows: a diagnosis of MS, age > 18 years, NTZ treatment, an assessment of the baseline CSF < 6 months before NTZ initiation, a follow-up of the CSF performed during NTZ treatment < 24 months after NTZ initiation and a follow-up of the CSF performed < 6 weeks after NTZ discontinuation. We excluded patients who had received high dose steroids < 30 days before the CSF assessment, had a relapse activity < 30 days before the CSF, were pregnant or had blood contamination of the CSF (number of erythrocytes > 50/mcL) (Figure 1) [23]. The study protocol was approved by the local ethics board. All patients gave their written informed consent. The study was carried out according to the Declaration of Helsinki and guidelines for good clinical practice.

### 2.2. Observation and Therapy

The study included clinical visits every 3–6 months during the follow-up in routine clinical practice. All patients started the treatment at the baseline with 300 mg of intravenous NTZ every 4 weeks (Tysabri^®^, Biogen-Idec, Cambridge, MA, USA) during the first year and every 4–6 weeks afterwards. Relapses during the study were treated with 3–6 g of methylprednisolone.

### 2.3. MRI Acquisition and Analysis

All MRI scans were performed with a standardized protocol at a central site on the same 1.5 T scanner (Gyroscan; Philips Medical Systems, Best, The Netherlands). The MRI scans were performed at least 30 days after intravenous high dose steroids and not more than 120 days after or before their respective spinal tap (Figure 2). Axial brain acquisitions included fluid attenuated inversion recovery (FLAIR) and 3D T1-weighted images (3D-T1W). Image analyses included the analyses of changes in T2 lesions and whole brain volumes. Absolute T2 lesion volumes and brain parenchymal fractions were obtained using ScanView software [24] and percent changes in whole brain (WB) volumes were obtained using the SIENA method [25,26]. Lesion filling on 3D T1-WI images was used to reduce the impact of T1 hypo-intensities on tissue segmentation [27].

### 2.4. Lumbar Puncture and Cerebrospinal Fluid Examination

All lumbar punctures were performed at the General University Hospital in Prague, Czech Republic. The mean time between the baseline CSF and NTZ initiation was 7.7 weeks (SD 5.6 weeks; median 6.1; range 0.3–26.7 weeks). The mean time between the follow-up CSF and NTZ initiation was 151.4 weeks (SD 92.7 weeks; median 119.6; range 40–567.4 weeks). A total of 24 patients had a follow-up CSF after NTZ discontinuation (mean 4.4 weeks). For the biochemical assessments, the CSF was drawn from the L5–S1, L4–5 or L3–4 interspace in an upright sitting or lying position in the morning hours using a standard sterile preparation and a 20G Sprotte atraumatic needle. A total of 10–15 mL of CSF and a 5 mL volume of blood were obtained. The total protein in the CSF was determined photometrically on a Beckman Coulter analyzer (Synchron LX 20) with a reaction with pyrogallol redmolybdate. Albumin, immunoglobulin G (IgG) and immunoglobulin M (IgM) concentrations were quantified in serum and the CSF by immunonephelometry on a Beckman Coulter analyzer (IMMAGE). The QAlb was defined as the ratio of the albumin concentration in the CSF divided by the albumin concentration in the blood serum (QAlb = (albumin in CSF (mg/L)/albumin in serum (mg/L))*1000). The identification of OCBs was performed by isoelectric focusing (IEF) with an ultra-sensitive immunofixation on Sebia (Hydrasys Focusing). All CSF examinations were analyzed in the same laboratory with identical procedures.

### 2.5. Statistical Analysis

All analyses were performed using R statistical software available online: http://www.R-project.org (accessed on 26 November 2021) and SPSS 22.0 (IBM, Armonk, NY, USA). The normality of the distribution was assessed using the Kolmogorov–Smirnov method and a visual inspection of the histograms. Non-normally distributed variables were (log x + 1) transformed. The changes of brain MRI volumetric measures were annualized to adjust for different times between the MRI timepoints.

The associations among the CSF and other clinical and paraclinical measures were analyzed using Spearman’s correlation. Considering repeated measurements on a single patient, the MRI measures at different timepoints within patients were compared using linear mixed models (lme4 package; version 1.1-27.1) [28] with a fixed slope and random intercept specified for each patient and adjusted for age at the baseline exam, years between the CSF exams, expanded disability status scale (EDSS) at the baseline and level of the CSF measure at the baseline. Details about the structure of the statistical models are provided in Appendix A. Variance inflation factors (VIF) were used to examine multicollinearity among the independent variables (cut-off < 5) [29].

A *p*-value < 0.05 was considered to be statistically significant, indicating that there was less than a 5% probability that the null hypothesis was correct. The Benjamini–Hochberg (BH) procedure with *p* < 0.05 was used to control the false discovery rate (FDR) when conducting multiple comparisons (the number in (brackets) refers to the number of comparisons). The *p*-values from the multivariate models were not corrected for the FDR.

## 3. Results

### 3.1. Descriptive Characteristics of the Sample

We analyzed 186 paired CSF measurements from 93 RRMS patients. The average time between the baseline and follow-up spinal taps was 3.1 years (interquartile range 2.0–3.4; min–max: 1.0–11.4). The majority of patients were treated with low efficacy DMT before NTZ initiation (59 of 93; 63.4%). Table 1 describes the detailed characteristics of the patients at the baseline and at the follow-up.

### 3.2. Correlation between the CSF and Other Clinical and MRI Measures at the Baseline

Patients with a younger age (B = −0.35, *p* = 0.008), a shorter disease duration (B = −0.442, *p* < 0.001), a lower EDSS (B = −0.413, *p* < 0.001) and a higher BPF (B = 0.519, *p* < 0.001) had a higher number of CSF leukocytes (Appendix A). In a multivariable model of 64 patients (29 patients without MRI data), a shorter disease duration (*p* = 0.027), a lower EDSS (*p* = 0.041) and a higher BPF (*p* = 0.028) were independent correlates of a higher number of CSF leukocytes at the baseline.

### 3.3. Evolution of CSF Measures over the Follow-Up

NTZ treatment induced a decrease of CSF leukocytes (mean: 4.45 vs. 0.93 in mcL), CSF protein levels (0.34 vs. 0.31 mg/L) and albumin quotients (4.76 vs. 4.34). In addition, NTZ also decreased the IgG and IgM quotients and indices and decreased the number of CSF-restricted OCBs (9.91 vs. 7.91; all < 0.008) (Table 1; Figure 3).

In 9 (9.7%) patients, the OCBs disappeared (the positivity dropped from 94.6% to 86%). These patients had a lower IgG index at the baseline and after the follow-up; they did not differ in clinical or radiological disease activity at the baseline or over the follow-up compared with the other patients. On the other hand, in 26 (28%) patients, the number of OCBs increased over the follow-up. However, patients with an increase in OCBs did not differ in the basic characteristics in comparison with those with a stable or decreased number of OCBs at the follow-up (data not shown).

### 3.4. Predictors of the Evolution of CSF Measures over the Follow-Up

A higher age at the baseline (B = 0.02, *p* = 0.032) and a longer time on the NTZ treatment (B = 0.13, *p* = 0.004) were associated with a higher decrease of an NTZ-induced reduction of the albumin quotient. Otherwise, only the baseline levels of different CSF measures were associated with an extent of the decrease of CSF inflammatory measures during the NTZ treatment. Other variables such as age, disease duration, EDSS or MRI indices were not associated with changes of CSF measures following NTZ initiation (Appendix A).

### 3.5. Evolution of the Clinical and MRI Measures over the Follow-Up

We found a reduction in the relapsing activity and stable EDSS over the follow-up (Table 1). We did not find a difference in the T2-LV (*n* = 45) and BPF (*n* = 64) between the baseline and the follow-up (B = 0.00, *p* = 0.99 and B = −0.08, *p* = 0.40, respectively) (Table 1). The average annualized brain volume loss during the follow-up was −0.50% (IQR = −0.96, −0.16) and was predicted by the baseline IgM index (B = −0.46; *p* = 0.001; adjusted-p(5) = 0.005) but not by the baseline IgG (B = −0.32; *p* = 0.10). In other words, a 10% higher level of the IgM index at the baseline was associated with a 0.046 greater annualized brain volume loss over time (Figure 4). The predictive value of the baseline CSF leukocytes for the evolution of brain volume loss was lost after a correction for the FDR (B = −0.31; *p* = 0.034; adjusted-p(5) = 0.10). The predictive values of the IgG and IgM indices also remained significant in the multivariable models adjusted for the baseline CSF leukocytes. An annualized change of T2-LV over the follow-up was not predicted by any baseline CSF measures (Appendix A).

## 4. Discussion

In this study, NTZ induced a reduction of a wide spectrum of basic CSF inflammatory measures such as the number of CSF leukocytes, CSF proteins, albumin quotients and IgG and IgM quotients and indices. Overall, NTZ led to the normalization of CSF measures in the majority of patients and, hence, the most obvious treatment effect on the biochemical parameters was observed in patients with the highest degree of inflammation before NTZ treatment initiation. Indeed, patients with a younger age, a shorter disease duration, a lower EDSS and a higher BPF had a higher number of CSF leukocytes at the baseline. This was consistent with recent research showing a negative correlation between a younger age and higher CSF leukocytes [30] but also with a number of studies documenting a higher inflammatory activity in young patients and in early disease stages [25,31,32]. Reductions in the IgG and IgM indices and quotients following NTZ have also been reported previously [13,15,16]. However, only the study by Schluter et al. found that NTZ induced a reduction of the IgG index but not the IgM indices in RRMS patients [13].

Although the number of OCBs reduced after NTZ treatment, OCBs disappeared completely only in 9 patients (9.7%) and OCBs positivity dropped from 94.6% to 86.0%. This was consistent with previous studies reporting a decrease in OCB positivity from a 100% positivity at the baseline to an 84% positivity at the follow-up (*n* = 73) [14], from an 86% positivity at the baseline to a 76% positivity at the follow-up (*n* = 33 before and 49 after NTZ treatment) [13] and a 93.3% positivity at the baseline to an 80% positivity at the follow-up (*n* = 15) [15]. Only one study by Mancuso et al. observed a complete disappearance of OCBs in the majority of patients on NTZ with a decrease in the OCB positivity from 92% at the baseline to 42% at the follow-up (*n* = 24) [17]. Interestingly, in 26 (28%) of our patients, there was an increase in the number of CSF-restricted OCBs independent to the reduction of CSF leukocytes, Ig production or albumin quotients. These findings may indicate that OCBs (mostly memory IgG) reflect previous rather than ongoing neuroinflammation [13,33]. Patients in our study whose OCBs disappeared or increased did not differ from the rest of our sample in terms of their baseline characteristics or disease activity over the follow-up.

Supporting previous research, we also found that NTZ treatment was also associated with relatively stable clinical (EDSS) and radiological (T2-LV and BPF) measures over the follow-up [1,2,3,4,34]. Given that we analyzed the brain volumes after a longer follow-up period (mean 2.4 years), we did not capture the early acceleration of the brain volume loss attributed to the pseudoatrophy effect. In this respect, several previous studies that showed an accelerated brain volume loss early after NTZ treatment initiation may not be in contradiction with our results [35,36,37]. Interestingly, our results found that the average annualized brain volume loss during the follow-up was predicted by the IgM index at the baseline, supporting the role of intrathecal IgM production for the prediction of future disease activity [38,39].

Taken together, our results were in agreement with previous research showing the strong anti-inflammatory properties of NTZ. Given that NTZ targets mostly short-lived B cells (plasmablasts) rather than long-lived plasma cells (Schluter et al., 2021 [13]), NTZ treatment—as expected—resulted in a decrease of the inflammatory cell levels in the CSF, an improvement of the BBB permeability and a decrease of the intrathecal Ig levels. The association of the intrathecal IgM production with the subsequent inflammatory-driven brain volume loss was also in line with these findings. On the other hand, in the majority of patients, NTZ was associated with a relative persistence of the number of OCBs produced by long-lived plasma cells.

Although NTZ is a highly effective and safe DMT, the risk of progressive multifocal leukoencephalopathy (PML) in human polyomavirus JC virus (JCV)-positive patients represents the most important drawback responsible for not initiating or discontinuing NTZ in a great proportion of patients (Schwab et al., 2017 [40]). Hence, future multicenter studies on larger samples are needed to investigate whether there are any subgroups of patients with different safety profiles regarding JCV positivity and PML risk. We also need to understand the effect of NTZ in patients with progressive MS phenotypes or in advanced disease stages. There are a few limitations to this study. Our analyses did not include more detailed immunological markers such as lymphocyte subpopulations or cytokines, which may limit further insights into the effects of NTZ. The analysis of IgA, IgG and IgM OCBs was performed only for a few patients. Only a few of the patients had available MRI data fulfilling the inclusion criteria. On the other hand, our study had a moderate sample size of 93 patients with good quality clinical data. All patients underwent a uniform MRI protocol with MRIs performed using the same scanner.

In conclusion, we demonstrated that NTZ treatment induced a significant reduction of a spectrum of CSF inflammatory measures and the stabilization of radiological disease activity. The anti-inflammatory effect of NTZ was most evident in younger patients and in those at the early stage of the disease who had the highest inflammatory activity. Our findings emphasize the need for effective treatment strategies in early disease stages when inflammatory disease activity tends to be the highest.

## Figures and Tables

**Figure 1 diagnostics-11-02230-f001:**
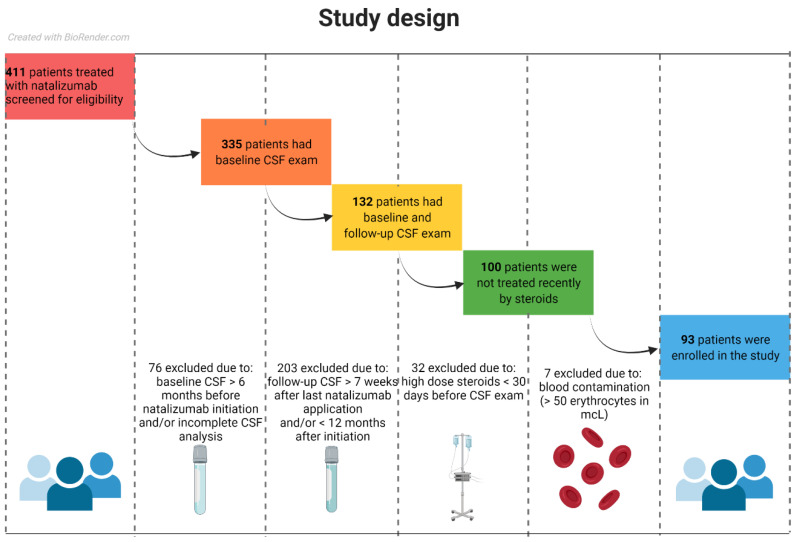
The study design. Originally, 411 patients who had received NTZ therapy were screened for their eligibility for the study. Exclusion criteria were applied to filter patients who had valid baseline and follow-up CSF data, had no history of recent high dose steroids and had CSF data without contamination by erythrocytes. Together, 93 patients fulfilled the inclusion criteria and were enrolled in the study.

**Figure 2 diagnostics-11-02230-f002:**
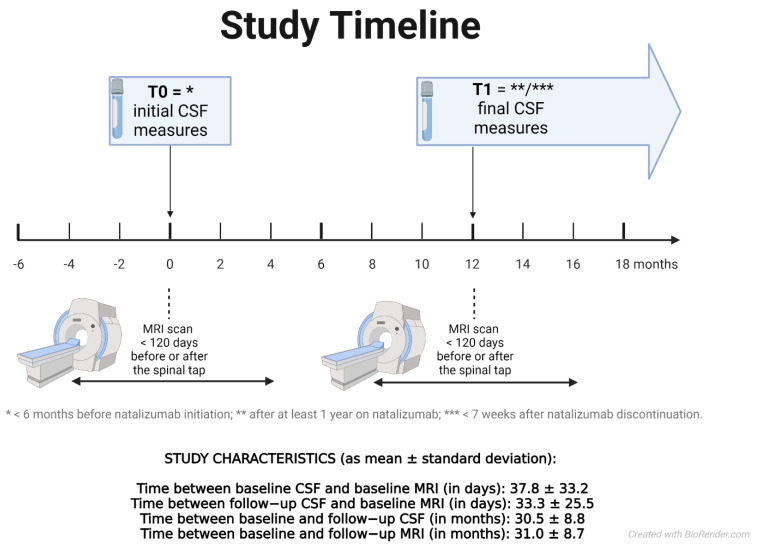
The study timeline. The baseline CSF exam for each patient was less than 6 months before NTZ initiation to ensure that it reflected the actual pre−treatment disease activity. The follow−up CSF exam for each patient was recorded at least 12 months after the initiation of therapy to ensure a sufficient exposition to the NTZ treatment and to ensure sufficient time for changes of biochemical measures. Each of the MRI scans (for the baseline and follow−up timepoint) were performed within 120 days before or after the respective CSF exam to accurately represent the concurrent radiological disease burden. If the NTZ treatment was discontinued in a patient, the final CSF data were recorded a maximum of 7 weeks after the discontinuation of the therapy to rule out a rebound effect.

**Figure 3 diagnostics-11-02230-f003:**
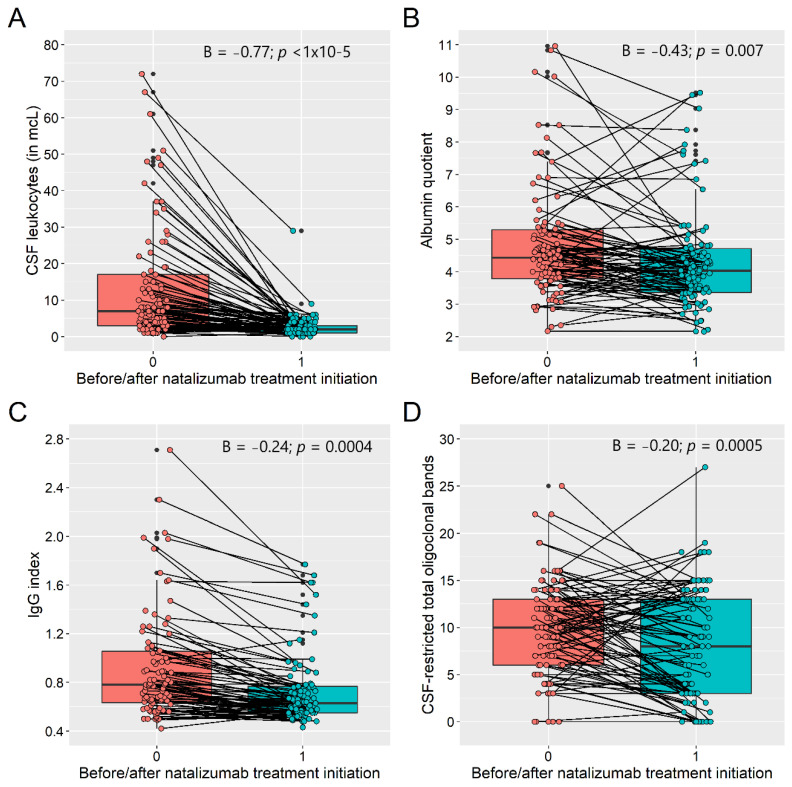
Natalizumab−induced changes of CSF leukocytes (**A**); albumin quotient (**B**); IgG index (**C**); and CSF-restricted oligoclonal bands (**D**).

**Figure 4 diagnostics-11-02230-f004:**
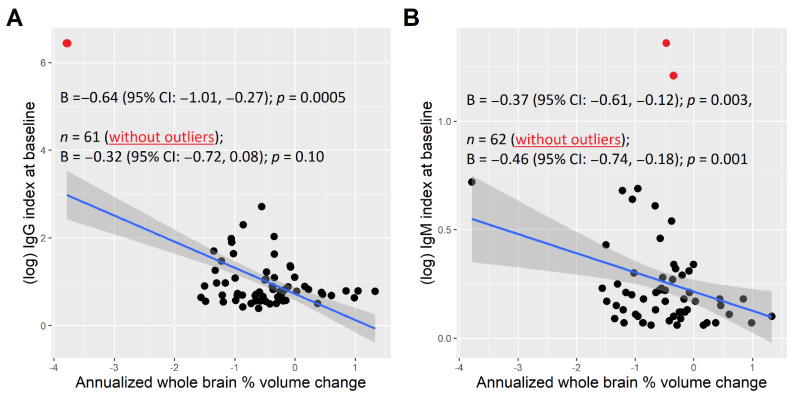
Association between the baseline IgG (**A**) and IgM indices (**B**) and brain % volume loss over the follow−up. Shown are the regression lines (in blue) with the confidence intervals (in grey). The values of the IgG/M indices greater or less than 2.0 standard deviations from the mean were defined as the outliers (in red). A multivariable mixed model with a fixed slope and random intercept specified for each patient adjusted for age and disease duration at the baseline exam, EDSS at the baseline exam and years between the baseline and follow-up timepoint.

**Table 1 diagnostics-11-02230-t001:** Baseline and follow-up characteristics of the 93 relapsing-remitting MS patients treated with natalizumab.

Patient Characteristics	Baseline	Follow-Up	Statistical Comparison	
			β (95% CI), *p*-Value	FDR *p*-Value ****
Demographic and clinical:				
Follow-up duration (years)	--	3.1, 2.4 (2.1–3.4)	--	--
Number of females	62 (67%)	--	--	--
Age (years)	35.21, 35.30, 12.8 (29.2–42.0)	38.26, 39.0, 11.6 (32.3–43.9)	--	--
Disease duration (years)	9.03, 7.94, 10.4 (3.3–13.7)	12.08, 11.1, 9.8 (7.2–17.0)	--	--
EDSS	3.35, 3.5, 1.5 (2.5–4.0)	3.31, 3.5, 2.0 (2.0–4.0)	−0.04 (−0.19; 0.11), *p* = 0.57	--
Relapses during last 12 months *	1.67, 2.0, 1 (1–2)	0.28, 0.0, 0 (0–0)	−4.0 (−5.15; −3.06), *p* = 6 × 10^−15^ ***	--
DMT **	33.3% no 63.4% low efficacy 3.2% moderate/high efficacy	--	--	--
Natalizumab treatment (years)	--	5.53, 4.53, 6.7 (2.3–9.0)	--	--
CSF measures:				
CSF leukocytes (mcL) ^†^	4.45, 2.33, 4.7 (1.0–5.7)	0.93, 0.67, 0.7 (0.3–1.0)	−0.77 (−0.93; −0.60), *p* < 1 × 10^−15^	<0.0001
CSF total protein (mg/L)	0.34, 0.31, 0.12 (0.27–0.39)	0.31, 0.3, 0.09 (0.24–0.33)	−0.03 (−0.04; −0.01), *p* < 0.00007	0.0002
CSF total albumin (mg/L)	0.20, 0.18, 0.07 (0.15–0.22)	0.19, 0.18, 0.06 (0.15–0.21)	−0.01 (−0.02; 0.00), *p* = 0.10	NS
QAlb	4.76, 4.41, 1.6 (3.7–5.3)	4.34, 4.03, 1.3 (3.4–4.7)	−0.43 (−0.74; −0.11), *p* = 0.007	0.008
Abnormally increased QAlb	10 from 93 (11%)	5 from 93 (5%)	^‡^	--
CSF-restricted total OCB	9.91, 10.0, 7.5 (6.0–13.5)	7.91, 8.0, 10 (3.0–13.0)	−2.0 (−3.2; −0.9), *p* = 0.0005	0.0007
CSF-restricted OCB positivity	88 from 93 (95%)	79 from 93 (85%)	^‡^	--
IgG quotient (*n* = 90)	4.45, 3.83, 2.3 (2.6–4.9)	3.06, 2.67, 1.68 (2.09–3.77)	−1.21 (−1.51; −0.90), *p* = 6 × 10^−15^	<0.0001
IgM quotient ^†^ (*n* = 80)	1.06, 0.8, 0.78 (0.52–1.3)	0.81, 0.65, 0.48 (0.42–1.0)	−0.12 (−0.19; −0.06), *p* = 0.0002	0.0004
IgG index (*n* = 90)	0.97, 0.78, 0.43 (0.65–1.08)	0.73, 0.63, 0.22 (0.55–0.77)	−0.24 (−0.38; −0.11), *p* = 0.0004	0.0007
Abnormally increased IgG index	64 from 92 (69%)	39 from 87 (45%)	−14.4 (−20.4; −10.8), *p* = 1 × 10^−12^ ***	<0.0001
IgM index ^†^ (*n* = 80)	0.24, 0.18, 0.2 (0.1–0.3)	0.19, 0.14, 0.13 (0.09–0.22)	−0.04 (−0.06; −0.09), *p* = 0.003	0.004
MRI measures:				
T2-LV (mL) (*n*1 = 64; and *n*2 = 45) ^†^	7.1, 3.1, 9.0 (1.5–10.5)	7.9, 4.0, 8.7 (1.8–10.5)	0.00 (−0.08; 0.08), *p* = 0.99	NS
BPF (%) (*n* = 78)	84.4, 84.6, 3.2 (83.0–86.2)	84.35, 84.54, 3.4 (82.7–86.1)	−0.08 (−0.29; 0.11), *p* = 0.40	NS

β = estimate of the linear mixed model; BPF = brain parenchymal fraction; CI = confidence interval; CSF = cerebrospinal fluid; DMT = disease modifying therapy; EDSS = expanded disability status scale; IgG = immunoglobulin G; IgM = immunoglobulin M; *n* = number at the baseline and follow-up; *n*1 = number of available measures at the baseline; *n*2 = number of available measures at the follow-up; NS = not statistically significant; OCB = number of oligoclonal bands; QAlb = albumin quotient; T2-LV = T2 lesion volume. Unless otherwise indicated, all data are reported as mean, median and interquartile ranges. Differences between the groups were tested using a mixed model analysis with a fixed slope and random intercept specified for each patient adjusted for age at the baseline exam, time between the CSF exams (in years) and EDSS at the baseline exam. Presented estimates of the models reflect an association between the CSF outcome measures (dependent variable) and the timepoint (before or after natalizumab treatment initiation) treated as a binary categorical independent variable. * from the CSF exam. ** low efficacy DMT: glatiramer-acetate and interferons; moderate/high efficacy DMT: cladribine, fingolimod, mitoxantrone. *** reported exponentially transformed β representing odds ratios (βOR). **** *p*-values adjusted for the false discovery rate (FDR) by the Benjamini–Hochberg procedure. The number of comparisons was 10. ^†^ [log + 1] transformed variable. ^‡^ not analyzed due to small sample sizes of patient subgroups. Albumin quotient (QAlb) = (albumin in CSF (mg/L)/albumin in serum (mg/L)) × 1000. Abnormally increased QAlb if: >6.5 if age < 40 years; or >8.0 if age > 40 years. IgG (or IgM) quotient = (IgG in CSF (mg/L)/IgG in serum (mg/L)) × 1000. IgG (or IgM) index = (IgG in CSF/IgG in serum)/(albumin in CSF/albumin in serum); all in mg/L. Abnormally increased IgG index if: >0.65.

## Data Availability

Anonymized data not published within this article may be made available upon request from a qualified investigator.

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
