# Peer review of "Natalizumab Induces Changes of Cerebrospinal Fluid Measures in Multiple Sclerosis"

_diagnostics, 2021, doi:10.3390/diagnostics11122230_

Round 1

Reviewer 1 Report

The authors have evaluated the changes induced in CSF after Natalizumab therapy in MPS patients. The topic is of interest for the MS research field, however, the manuscript must be significantly improved before acceptance. Additionally, the English must extensively be revised.

Firstly, in the abstract/methods:

"The associations between CSF and..." in terms of what was analyzed? causality, correlation? is a too convoluted formulation, please rephrase in order to make the statement more clear.

The concluding remark that "NTZ is associated with a major reduction..." does the major reduction refer to the statistical significance of the findings, or is it in terms of absolute concentration levels detected in the CSF?

Hypothesis testing: For the sake of clarity, please state in the manuscript both the research and the statistical null hypotheses. What was the considered significance level to reject H0?

The clear hypothesis of the study must be highlighted. 

Statistical model: Besides mentioning statistical software packages, there is no reference to any textbook / technical paper about the statistical models and methods deployed in the manuscript. The interested reader is left with no possibility to reproduce or apply the same method to their own research. In the linear mixed models deployed, the intercepts were random. What about the slopes, there is nothing mentioned about them. Correlation coefficients are not shown for any of the performed regressions. In the supplementary material (e.g. Table 2), in terms of independent variables, except for the NTZ treatment and baselines of the dependent variables in question, there seems not to be much of a statistical significance for the other independent variables, with maybe a few exceptions. Was it necessary to crunch all the listed independent variables through the mixed model? Before applying the mixed model, have you considered screening for variables with the largest variance by means of e.g. dimensionality reduction when addressing predictor collinearity? How did you establish the cut-off values for the calculated VIFs?

Key findings of previous work, and how it relates to own work:

Although extensively referring to other studies throughout the text and critically assessing against own findings, a literature review paragraph with key findings and a framing into a broader picture with regard to the status-quo of the subject and applied methods is desirable.

Given the number of abbreviations, it is rather tedious to read the manuscript, where the reader is sent to search for the first occurrence within the text, where the meaning is defined. Unless restricted by the publisher, it is recommended that a nomenclature be included at the end of the manuscript.

Citations: Put all citations before the ending point of a sentence. As it currently is, it’s visually challenging to locate the beginning of a new sentence throughout the text as well as a corresponding reference.

Other aspects:

-Hematopoietic stem and progenitor cells are blood and not serum biomarkers.

- In the paragraph describing the association between NTZ and biomarkers, It would be better if the authors not only enumerate all these markers but also provide a short insight into their role in MS pathophysiology.

- Please provide the study protocol approval number and the statement if the study was conducted according to the Helsinki declaration.

- Include in timeline from figure 2 all relevant events for the study, also clearly differentiate between CSF/NTZ and follow-up CSF and NTZ. How come there are exactly the same values for the two distributions. 
Clarify why the arithmetic mean is negative (-28.2) weeks, while the range is between 0  and 103 weeks.

- The formula for Qalb is incorrect

- Results section: it is not possible to understand the total duration and structure of the study based on the information provided in the text and the schematic representations from figure 2. Please provide a complete representation of the timeline valid for the complete duration of the study where you include all key events/ interactions with patients.

- The last paragraph at the end of section 3: this cumbersome paragraph needs complete reformulation.

-For Section 5 Evolution of clinical and MRI measures: it's understood that the blue curve and its CI correspond to regression performed on the dataset without outliers. what was the exclusion criterion for the outliers?
please comment. What about the resulting correlation coefficient of the regression? can you communicate its values?

-Page 7: line "86% to 76% decrease ...." please provide details, not a clear statement.

Table 1: Please provide the appropriate assignment for all listed terms within the header/first column, with a corresponding explanatory footnote or within the below existing legend, in other words, fully describe the content of the table.

Supplementary Table 1: What does stand for in the table? There is no explanation in the Legend. Note that commonly adopted convention for reporting p-values assigns stars as follows: p-value < 0.05 , p-value < 0.01 , p-value < 0.001 ***. 

Line 4 Table 1 suppl: please correct Albumine.

Major revision required: please address the above specified unclarified points, topics.

Author Response

Reviewer: 1

The authors have evaluated the changes induced in CSF after Natalizumab therapy in MPS patients. The topic is of interest for the MS research field, however, the manuscript must be significantly improved before acceptance. Additionally, the English must extensively be revised.

Comment: *1. Firstly, in the abstract/methods: "The associations between CSF and..." in terms of what was analyzed? causality, correlation? is a too convoluted formulation, please rephrase in order to make the statement more clear.

Answer: We agree with this comment. The statement was modified.

„The effect of Natalizumab on CSF, clinical and paraclinical measures was analyzed using adjusted mixed models.“

Comment: *2. The concluding remark that "NTZ is associated with a major reduction..." does the major reduction refer to the statistical significance of the findings, or is it in terms of absolute concentration levels detected in the CSF?

Answer: To not confuse readers, we deleted the word „major“ which does not refer to any statistical result.

Comment: *3. Hypothesis testing: For the sake of clarity, please state in the manuscript both the research and the statistical null hypotheses. What was the considered significance level to reject H0?

Answer: This was added in the introduction and in the statistical methods section. A p-value <0.05 was considered as statistically significant indicating that there is less than a 5% probability the null hypothesis is correct.

Comment: *4. The clear hypothesis of the study must be highlighted.

Answer: This was added in the introduction.

Comment: *5. Statistical model: Besides mentioning statistical software packages, there is no reference to any textbook / technical paper about the statistical models and methods deployed in the manuscript. The interested reader is left with no possibility to reproduce or apply the same method to their own research.

Answer: We agree with the reviewer´s comment and added references to statistical software package applied for statistical analysis. Details about dependent and independent variables are described in Table footnotes and Supplementary Table 2.

Comment: *6. In the linear mixed models deployed, the intercepts were random. What about the slopes, there is nothing mentioned about them.

Answer: Given that we did not expect that response of NTZ treatment should vary by subject, we used fixed slopes. This is now mentioned in the text.

Comment: *7. Correlation coefficients are not shown for any of the performed regressions.

Answer: We added the correlation and regression coefficients through the manuscript.

Comment: *8. In the supplementary material (e.g. Table 2), in terms of independent variables, except for the NTZ treatment and baselines of the dependent variables in question, there seems not to be much of a statistical significance for the other independent variables, with maybe a few exceptions. Was it necessary to crunch all the listed independent variables through the mixed model? Before applying the mixed model, have you considered screening for variables with the largest variance by means of e.g. dimensionality reduction when addressing predictor collinearity? How did you establish the cut-off values for the calculated VIFs?

Answer: We thank reviewer for this important comment. Although, the most precise definition of covariates is by an analysis of covariance, in our study the covariates selection was guided by background knowledge. The models presented in Table 1 were adjusted for age, EDSS at baseline and time between two timepoints (e.g., years between baseline and follow-up).

In addition to these primary results, in the Supplementry Table 2 we showed models adjusted also for disease duration at baseline and baseline levels of CSF measures. Although, many of these covariates were not significant, we kept them in the models to emphasize (better illustrate) their limited effect on the evolution of biochemical measures after treatment intervention. If recommended by the reviewer, we may delete Supplementary Table 2.

Regarding VIF, we used conservative cut-off value of <5.0 (Hair et al., 1995; Ringle et al., 2015)

References:

-Hair, J. F. Jr., Anderson, R. E., Tatham, R. L. & Black, W. C. (1995). Multivariate Data Analysis (3rd ed). New York: Macmillan.

-Ringle, Christian M., Wende, Sven, & Becker, Jan-Michael. (2015). SmartPLS 3. Bönningstedt: SmartPLS. Retrieved from http://www.smartpls.com

Comment: *9. Key findings of previous work, and how it relates to own work: Although extensively referring to other studies throughout the text and critically assessing against own findings, a literature review paragraph with key findings and a framing into a broader picture with regard to the status-quo of the subject and applied methods is desirable.

Answer: This was added in the discussion.

Comment: *10. Given the number of abbreviations, it is rather tedious to read the manuscript, where the reader is sent to search for the first occurrence within the text, where the meaning is defined. Unless restricted by the publisher, it is recommended that a nomenclature be included at the end of the manuscript.

Answer: This was added at the end of the discussion.

Comment: *11. Citations: Put all citations before the ending point of a sentence. As it currently is, it’s visually challenging to locate the beginning of a new sentence throughout the text as well as a corresponding reference.

Answer: This was checked and corrected if applicable.

Comment: *12. Other aspects: Hematopoietic stem and progenitor cells are blood and not serum biomarkers.

Answer: This was corrected.

Comment: *13. In the paragraph describing the association between NTZ and biomarkers, It would be better if the authors not only enumerate all these markers but also provide a short insight into their role in MS pathophysiology.

Answer: We added additional description of the effect of natalizumab on T and B-cells.

All these results indicate immunomodulatory effects of NTZ (Meinl et al., 2006), which besides modulating of T-cell migration, is strongly affecting the B-cell compartment by inhibiting anti-a4b1 B-cell migration across the blood–brain barrier (Alter et al., 2003), by participating in activation of human memory B cells (Silvy et al., 1997), and by supporting the long-lived plasma cells to remain in their survival niche (Shapiro-Shelef et al., 2005).

References:

-Meinl, E., M. Krumbholz, and R. Hohlfeld, B lineage cells in the inflammatory central nervous system environment: migration, maintenance, local antibody production, and therapeutic modulation. Ann Neurol, 2006. 59(6): p. 880-92.

-Alter, A., et al., Determinants of human B cell migration across brain endothelial cells. J Immunol, 2003. 170(9): p. 4497-505.

-Silvy, A., et al., A role for the VLA-4 integrin in the activation of human memory B cells. Eur J Immunol, 1997. 27(11): p. 2757-64.

-Shapiro-Shelef, M. and K. Calame, Regulation of plasma-cell development. Nat Rev Immunol, 2005. 5(3): p. 230-42.

Comment: *14. Please provide the study protocol approval number and the statement if the study was conducted according to the Helsinki declaration.

Answer: This was added.

Comment: *15. Include in timeline from figure 2 all relevant events for the study, also clearly differentiate between CSF/NTZ and follow-up CSF and NTZ. How come there are exactly the same values for the two distributions.

Clarify why the arithmetic mean is negative (-28.2) weeks, while the range is between 0 and 103 weeks.

Answer: We appologize for non-clarity and mistakes in the results section. All results were checked and corrected. The figure 2 provides major timepoints of the study. The details are provided in the results. We hope that we have understod the reviewer´s comment properly.

Comment: *16. The formula for Qalb is incorrect

Answer: The formula for the Qalb was modified. In the previosu version we did not multiply by 100 but used different units for albumin in serum (mg/l instead of g/L).

QAlb = albumin in CSF (mg/L) / albumin in serum (mg/L)*1000

Comment: *17. Results section: it is not possible to understand the total duration and structure of the study based on the information provided in the text and the schematic representations from figure 2. Please provide a complete representation of the timeline valid for the complete duration of the study where you include all key events/ interactions with patients.

Answer: We updated the Figure 2 and provided a complete representation of the timeline.

Comment: *18. The last paragraph at the end of section 3: this cumbersome paragraph needs complete reformulation.

Answer: The last paragraph pf the statistical section was modified.

Comment: *19. For Section 5 Evolution of clinical and MRI measures: it's understood that the blue curve and its CI correspond to regression performed on the dataset without outliers. what was the exclusion criterion for the outliers?

please comment. What about the resulting correlation coefficient of the regression? can you communicate its values?

Answer: Indeed, regression lines (in blue) with confidence intervals (in grey) were performed on all available data, including outliers. Values of IgG/M indices greater or less than 2.0 standard deviations from the mean were defined as outliers (in red). We modifed/expanded the the Figure legend.

Given that IgG and IgM indices were log transformed in multivariable model, for example 10% higher level of IgM index at baseline was associated with 0.046 (B=-0.46) greater annualized brain volume loss over time (Figure 4.)

Comment: *20. Page 7: line "86% to 76% decrease ...." please provide details, not a clear statement.

Answer: The details were added.

Comment: *21. Table 1: Please provide the appropriate assignment for all listed terms within the header/first column, with a corresponding explanatory footnote or within the below existing legend, in other words, fully describe the content of the table.

Answer: We thank reviewer for this comment. The additional explanations were added in the footnote.

Comment: *22. Supplementary Table 1: What does stand for in the table? There is no explanation in the Legend. Note that commonly adopted convention for reporting p-values assigns stars as follows: p-value < 0.05, p-value < 0.01 , p-value < 0.001 ***.

Answer: The legend of the table and reporting of p-values was checked and modifed if applicable.

Comment: *23. Line 4 Table 1 suppl: please correct Albumine.

Answer: This was corrected.

Comment: *24. Major revision required: please address the above specified unclarified points, topics.

Answer: We have tried to address all comment and revised English language.

Reviewer 2 Report

The study of Ranjani Ganapathy S et al. aims to evaluate the effect of Natalizumab treatment in cerebrospinal fluid (CSF) of multiple sclerosis patients. The study appears very clear and well done, being in line with the current research trend that is moving towards the evaluation of biological drugs. Furthermore, the exclusion of patients who had received high-dose steroids makes the study even more valid, avoiding the overlapping of anti-inflammatory activities that would not have been easily traceable. However, although the contents are well appreciated, the quality of their presentation should be improved. Below are my minor comments.

  • Standardize text formatting (in font and size) throughout the manuscript following MDPI guidelines
  • Standardize the references according to the MDPI style
  • Avoid underlining in the introduction
  • In the introduction, add some background about multiple sclerosis (just 4-5 lines) in order to contextualize the use of Natalizumab
  • In the methods section add the numbers of the subsections to the following subparagraphs (example 2.1 Study population etc.)
  • Add a short description for Figure 1 and Figure 2
  • Given that the authors recognize the limitations of their study, at the end of the discussion it would be interesting to elucidate possible future perspectives, highlighting the possible directions that the authors could take in further analyzing the properties of Natalizumab.

Author Response

Reviewer: 2

The study of Ranjani Ganapathy S et al. aims to evaluate the effect of Natalizumab treatment in cerebrospinal fluid (CSF) of multiple sclerosis patients. The study appears very clear and well done, being in line with the current research trend that is moving towards the evaluation of biological drugs. Furthermore, the exclusion of patients who had received high-dose steroids makes the study even more valid, avoiding the overlapping of anti-inflammatory activities that would not have been easily traceable. However, although the contents are well appreciated, the quality of their presentation should be improved. Below are my minor comments.

Comment: *1. Standardize text formatting (in font and size) throughout the manuscript following MDPI guidelines

Standardize the references according to the MDPI style

Avoid underlining in the introduction

Answer: This was corrected.

Comment: *2. In the introduction, add some background about multiple sclerosis (just 4-5 lines) in order to contextualize the use of Natalizumab

Answer: This was added.

Comment: *3. In the methods section add the numbers of the subsections to the following subparagraphs (example 2.1 Study population etc.)

Answer: This was added.

Comment: *4. Add a short description for Figure 1 and Figure 2

Answer: This was added.

Figure 1. Study design. Originally, 411 patients who had received NTZ therapy were screened if they are eligible for the study. Exclusion criteria were applied to filter for patients who had valid baseline and follow-up CSF data, without history of recent high dose steroids, and without contamination by erythrocytes. Together, 93 patients fulfilled the inclusion cri-teria and were enrolled in the study.

Figure 2. Study timeline: The baseline CSF exam for each patient was less than 6 months before NTZ initiation, to ensure that it reflects actual pre-treatment disease activity. The follow-up CSF exam for each patient was recorded at least 12 months after the initiation of therapy, to ensure sufficient exposition to NTZ treatment and to ensure sufficient time for changes of biochemical measures. Each of the MRI scans (for the baseline and follow-up timepoint) were performed within 120 days before or after the respective CSF exam, to accurately represent the concurrent radiological disease burden. If NTZ treatment had been discontinued in a patient, the final CSF data was recorded a maximum of 7 weeks after the discontinuation of therapy to rule-out rebound effect.

Comment: *5. Given that the authors recognize the limitations of their study, at the end of the discussion it would be interesting to elucidate possible future perspectives, highlighting the possible directions that the authors could take in further analyzing the properties of Natalizumab.

Answer: This was added.

Round 2

Reviewer 1 Report

The manuscript was revised according to the recommendations.